# Distinctive Toll-like Receptors Gene Expression and Glial Response in Different Brain Regions of Natural Scrapie

**DOI:** 10.3390/ijms23073579

**Published:** 2022-03-25

**Authors:** Mirta García-Martínez, Leonardo M. Cortez, Alicia Otero, Marina Betancor, Beatriz Serrano-Pérez, Rosa Bolea, Juan J. Badiola, María Carmen Garza

**Affiliations:** 1Centro de Encefalopatías y Enfermedades Transmisibles Emergentes, IA2, IIS Aragón, Universidad de Zaragoza, 50013 Zaragoza, Spain; mirta@unizar.es (M.G.-M.); mbetancorcaro@gmail.com (M.B.); rbolea@unizar.es (R.B.); badiola@unizar.es (J.J.B.); 2Department of Medicine and Centre for Prions and Protein Folding Diseases, University of Alberta, Edmonton, AB T6G 2G3, Canada; 3Agrotecnio-CERCA Center, Department of Animal Science, University of Lleida, 25198 Lleida, Spain; beatriz.serrano@udl.cat; 4Departamento de Anatomía e Histología Humanas, IIS Aragón, Universidad de Zaragoza, 50009 Zaragoza, Spain; mcgarza@unizar.es

**Keywords:** prion, scrapie, toll-like receptors, neuroinflammation, microglia, astrocytes

## Abstract

Prion diseases are chronic and fatal neurodegenerative diseases characterized by the accumulation of disease-specific prion protein (PrP^Sc^), spongiform changes, neuronal loss, and gliosis. Growing evidence shows that the neuroinflammatory response is a key component of prion diseases and contributes to neurodegeneration. Toll-like receptors (TLRs) have been proposed as important mediators of innate immune responses triggered in the central nervous system in other human neurodegenerative diseases, including Alzheimer’s disease, Parkinson’s disease, and amyotrophic lateral sclerosis. However, little is known about the role of TLRs in prion diseases, and their involvement in the neuropathology of natural scrapie has not been studied. We assessed the gene expression of ovine TLRs in four anatomically distinct brain regions in natural scrapie-infected sheep and evaluated the possible correlations between gene expression and the pathological hallmarks of prion disease. We observed significant changes in TLR expression in scrapie-infected sheep that correlate with the degree of spongiosis, PrP^Sc^ deposition, and gliosis in each of the regions studied. Remarkably, *TLR4* was the only gene upregulated in all regions, regardless of the severity of neuropathology. In the hippocampus, we observed milder neuropathology associated with a distinct TLR gene expression profile and the presence of a peculiar microglial morphology, called rod microglia, described here for the first time in the brain of scrapie-infected sheep. The concurrence of these features suggests partial neuroprotection of the hippocampus. Finally, a comparison of the findings in naturallyinfected sheep versus an ovinized mouse model (tg338 mice) revealed distinct patterns of TLRgene expression.

## 1. Introduction

Transmissible spongiform encephalopathies (TSEs), or prion diseases, are a group of fatal neurodegenerative disorders that include Creutzfeldt–Jakob disease (CJD) in humans, scrapie in sheep and goats, bovine spongiform encephalopathy (BSE) in cattle, and chronic wasting disease in cervids (CWD). These TSEs are characterized by the transformation of the host-encoded prion protein (PrP^C^) into an infectious isoform, PrP^Sc^, which preferentially accumulates in the nervous system and lymphoreticular system (LRS), and an inflammatory immune reaction in the central nervous system (CNS) [1,2].

Prion diseases, such as scrapie, can be naturally acquired by oral exposure to PrP^Sc^. After crossing the intestinal barrier and undergoing primary replication in lymphoid tissues, PrP^Sc^ reaches the CNS via a process known as neuroinvasion. The infection propagates from sites in the gastrointestinal tract via the vagus and splanchnic nerves to the brainstem and spinal cord, respectively [3,4]. On reaching the CNS, PrP^Sc^ spreads from the caudal to rostral regions, i.e., via the rostral brainstem to the cerebellum, diencephalon, and frontal cortex [5]. The anatomically stereotyped accumulation of PrP^Sc^ suggests that its spread is mediated by the neuronal connectivity and/or selective vulnerability of different brain regions. Host factors such as the PrP^Sc^ genotype, breed, and age at infection contribute to the magnitude of PrP^Sc^ deposition and the distribution of spongiform lesions. However, the molecular basis of the host–prion interactions that determine the selective targeting of PrP^Sc^ to specific brain regions remains unknown, and constitutes one of the most intriguing aspects of the prion strain phenomenon.

Originally, it was assumed that the immune system did not respond to PrP^Sc^ due to natural tolerance, but time-course studies revealed prompt activation and proliferation of microglia and astrocytes even before the onset of clinical signs and neuronal degeneration [6,7,8]. This early activation and proliferation of microglia and astrocytes appears to play a central role in the pathogenesis of prion disorders, although the molecular mechanisms involved in their activation and the precise role of both reactive astrocytes and microglia in the disease pathogenesis remains unclear [9,10]. Moreover, it is widely accepted that microglial and astroglial distribution and phenotype vary across brain regions and species [11,12,13,14], attributing intrinsic characteristics to each region that influence vulnerability to neurodegeneration [10,15,16]. This phenomenon is especially evident in the preclinical stage, during which both microglia and astroglia respond to prion infection in a region-specific manner. However, as the disease progresses, a more common neuroinflammatory response begins to predominate, diluting most of the regional characteristics [17,18].

Over the last few years, a role of Toll-like receptors (TLRs) in microglial and astroglial reactivity has been reported in several neurodegenerative disorders, underscoring their influence on the progression of these diseases [19,20,21]. However, data on the role of TLRs in prion diseases are scarce. A partially protective role of TLRs has been described in prion pathology. Specifically, it has been reported that the loss of *TLR2* [22] and *TLR4* [23] accelerates prion pathogenesis, while TLR9 stimulation slows disease progression in mouse models [24,25]. To date, most studies of TLRs in prion diseases have been performed in vitro or using mouse models [8,22,26,27,28,29], and their role in naturally infected animals and humans remains uncertain. To our knowledge, one study was performed in newborn lambs orally inoculated with scrapie (i.e., using a natural, but not naturally infected, host) and reported *TLR2* and *TLR4* overexpression in blood in clinical diseases stages [30]. Two studies have reported altered TLR gene expression in humans with sporadic CJD (sCJD) [31,32]. The dysregulation of genes involved in the immune response included the overexpression of *TLR4* and *TLR7* mRNA in the cerebellum and frontal cortex of sCJD MM1 patients, and of *TLR7* in the cerebellum of sCJD VV2 patients. Moreover, Areskeviciute et al. [32] included *TLR4* in the top 40 upstream regulators predicted to show differential expression in sCJD versus control brain samples.

We examined how natural scrapie infection influences the transcription levels of ovine TLRs and some cytokines in four brain regions with differing degrees of prion susceptibility, based on previous results [18]. Quantitative PCR (qPCR) analysis revealed changes in gene expression in the scrapie-infected group versus the control group and a brain region-dependent response. Interestingly, the hippocampus showed the most distinctive lesion profile, with specific reactive microglia morphology and *TLR* transcription pattern. 

Given the dearth of experimental data on TLR gene expression in mice overexpressing sheep PrP, we also performed experiments using tg338 mice intracerebrally inoculated with scrapie. Our results reveal some similarities with previous studies using wild type mice, but significant differences with respect to the findings in natural scrapie.

## 2. Results

### 2.1. Scrapie Neuropathology of Naturally Infected Sheep

We first confirmed the scrapie neuropathology in the naturally infected sheep included in this study, as previously described [33,34,35,36]. Spongiosis, PrP^Sc^ deposition, and reactive glia were examined in four regions of the infected and control sheep brains: the medulla oblongata (Mo), thalamus (Th), hippocampus (Hc), and frontal cortex (Fc). All naturally infected animals were in the clinical stage of the disease. However, the clinical status, based on the frequency and severity of the clinical signs, varied among sheep, ranging from incipient to advanced clinical stages (Table 1).

Figure 1 shows representative microphotographs for the assessment of vacuolation and immunostaining for PrP^Sc^, GFAP, and Iba1, taken in each of the selected regions in infected and control animals. As expected, no evidence of vacuolation or PrP^Sc^ deposition was detected in tissue sections from the control sheep. Conversely, in the scrapie-infected sheep, all neuroanatomic areas showed spongiosis, PrP^Sc^ deposition, and reactive glia (*p* < 0.05). 

The intensity of vacuolation, PrP^Sc^ deposition, and gliosis clearly decreased from the medulla oblongata to the frontal cortex. Interestingly, despite the close proximity of the thalamus and hippocampus, significant differences in the scores for all pathological markers were observed.

The greatest degree of spongiosis was observed in the medulla oblongata and the thalamus, in contrast to the very scattered vacuolation found in the hippocampus and frontal cortex. Intraneuronal vacuolation was abundant in the medulla oblongata nuclei, as shown in the dorsal motor nucleus of the vagus nerve, while neuropil spongiosis predominated in the thalamus and frontal cortex (Figure 1A). Very few vacuoles were found in the neuropil of the hippocampus.

All scrapie-infected sheep showed widespread PrP^Sc^ accumulation in the four CNS regions analyzed, except for three sheep that showed incipient clinical signs and were PrP^Sc^-negative by immunohistochemistry (IHC) and Western blot (WB) findings in the frontal cortex (Table 1). The intensity of PrP^Sc^ immunostaining revealed a caudal to rostral decrease, similar to that observed for spongiosis, with maximum intensity in the medulla oblongata and thalamus, the two regions with the greatest degree of vacuolation, and lower intensity in the hippocampus and frontal cortex (Figure 1B). In the hippocampus, PrP^Sc^ immunostaining showed a focal and uneven distribution, in contrast to the homogeneous, scattered punctate pattern observed in the frontal cortex sections. We observed four major morphological patterns of PrP^Sc^ deposition, as previously described [36]: two intracellular patterns, which included intraneuronal and intraglial (including intra-astrocytic and intra-microglial) deposition, and two cell membrane/extracellular patterns, which included neuropil-associated and glial cell-associated deposition (Appendix A). The intraneuronal and neuropil-associated patterns predominated in all brain areas. Remarkably, the intraglial and glial patterns were drastically reduced or even absent in the frontal cortex and hippocampus compared with the medulla oblongata and thalamus.

In parallel with IHC, we analyzed the presence of protease-resistant prions (PrP^res^) by WB. The brain samples from scrapie-infected sheep displayed the classical three-banded PrP^res^ pattern associated with classical scrapie, with a molecular mass of the non-glycosylated fragment of approximately 19 kDa (Figure 2). Despite a certain degree of variability, the PrP^res^ signal increased with the severity of the clinical signs in all brain regions and was more abundant in caudal brain areas. The WB for PrP^res^ was positive in the medulla oblongata and thalamus in all scrapie-infected sheep. By contrast, the hippocampus and frontal cortex samples were not positive for PrP^res^ in all infected animals. Although the IHC was positive for PrP^res^ in the hippocampus of all infected sheep, 4/13 were negative by WB. In the frontal cortex, 3/13 of sheep were PrP^res^-negative by IHC and 4/13 were PrP^res^-negative by WB.

In addition to vacuolation and PrP^Sc^ deposition, the activation of glial cells has been extensively documented as an early event in the pathogenesis of protein misfolding diseases. To characterize the reactive glia, all brain areas were subjected to IHC using anti-GFAP antibodies to detect astrocytes and anti-Iba1 antibodies to detect microglia.

Iba1 immunostaining, which detects both resting and activated microglia [37], was significantly increased in all brain regions in scrapie-infected versus control sheep, and revealed marked differences in the microglial morphology among brain regions in infected animals (Figure 1C). The activation status of the microglial cells is defined based on three main morphologies: ramified, hypertrophic, and ameboid [38,39]. Microglial cells in the “surveilling” or relatively inactive state have a ramified morphology, characterized by a small cell body with many fine processes, as observed in our control animals. The active state is characterized by an ameboid morphology, which is commonly associated with the phagocytic state [38], and consists of spherical, non-ramified cell bodies. The hypertrophic morphology is characterized by microglia with a larger cell body and shorter, thick processes, and is associated with the production and release of inflammatory mediators [40,41]. The ameboid morphology was the most abundant in the medulla oblongata of scrapie-infected sheep. In the thalamus, the ameboid morphology was observed, but did not predominate: a variety of large and thin processes intermingled with sparse, thick processes arising from elongated cell bodies, corresponding to hypertrophic microglia. The morphology of the proliferative microglia in the frontal cortex and hippocampus differed markedly to that of the medulla oblongata and thalamus. In both regions, hypertrophied microglia predominated and ameboid microglia were practically absent. Remarkably, the *Cornu Ammonis* 3 (CA3) region of the hippocampus showed the greatest differences in terms of microglia morphology, exhibiting a microglial pattern known as rod microglia that was not observed in any other brain region. These marked differences observed in the assessment of the hippocampus prompted us to perform a detailed examination of subfields CA1, 3, 4, and the *stratum lacunosum-moleculare* (SLM).

GFAP immunostaining revealed marked astrogliosis in scrapie-infected versus control sheep (Figure 1D). Morphologically, the astrocytes from infected sheep consistently displayed hypertrophic cell bodies and processes, confirming the presence of reactive astrogliosis, while control sheep showed the typical stellate morphology of the non-activated state. In the medulla oblongata and thalamus, GFAP immunostaining was more intense, with the appearance of astrocyte networks due to overlapping processes. In the hippocampus and frontal cortex, the astrocytes were evenly distributed, with the presence of isolated hypertrophic stellate bodies.

### 2.2. Neuropathological Features in Hippocampus of Sheep Naturally Infected with Scrapie

In the scrapie-infected and control sheep, the four following hippocampal areas were examined in greater detail for the presence of all pathological markers: the pyramidal cell layers CA1, CA3, and CA4, and the SLM (Figure 3).

Spongiosis in the hippocampus was very mild, and occasional vacuoles were observed in the CA1, CA3, and CA4, and in the SLM (Figure 3A). This limited neuronal loss contrasted with sparse, but focally intense PrP^Sc^ deposition in CA1 and CA3 (Figure 3B). Very mild PrP^Sc^ staining was observed in the CA4 and the SLM. Distinct PrP^Sc^ patterns were observed in these brain areas (Appendix A). Whereas intraneuronal and neuropil-associated PrP^Sc^ patterns predominated in the CA1 and CA3, intraglial deposition was the main morphological pattern observed in the SLM. Mild intraneuronal and neuropil-associated PrP^Sc^ patterns were consistently observed in the CA4, although the intensity was lower than that observed in CA1 and CA3.

Iba1 was constitutively expressed and sparsely distributed in the control samples. By contrast, marked microgliosis was observed in CA1, CA3, and CA4 and the SLM in scrapie-infected sheep. However, the microglial morphological features differed significantly between these brain regions. In the CA1 pyramidal layer, most microglial cells showed short processes with enlarged cell bodies, characteristic of the hypertrophic morphology. In CA4, the microglial cells were also hypertrophic, but displayed longer and thinner branches than the CA1 microglial cells. Meanwhile, the hypertrophic microglia in the SLM showed short and thin processes with swollen cellular bodies. By contrast, in CA3, we observed the aforementioned rod-microglia morphology [42]. These microglial cells had elongated cell bodies, with a long parallel primary process oriented perpendicular to the CA3 pyramidal cell layer, and subsidiary branches that were shorter than the long primary processes.

GFAP immunostaining was significantly increased in the CA1 and CA3 pyramidal cell layers from the scrapie-infected versus control sheep. Interestingly, the PrP^Sc^ scores were highest in CA1 and CA3. No differences were observed in the CA4 pyramidal cell layer or in the SLM.

### 2.3. Neuropathology of Scrapie-Infected tg338 Mice

The same neuropathological markers of spongiosis, PrP^Sc^ deposition, microgliosis, and astrogliosis analyzed in sheep were examined in the thalamus of clinically affected tg338 mice that had been intracerebrally infected with scrapie (Figure 4). Compared with the controls, significant differences (*p* < 0.05) in spongiosis were observed in the thalamus of scrapie-infected mice, in which the vacuoles were clearly restricted to the neuropil. PrP^Sc^ deposits were widely distributed at the level of the cytoplasm and neuropil. The assessment of gliosis revealed significant differences in Iba1 and GFAP immunostaining between the scrapie-infected and control mice, with both astrocytes and microglia displaying hypertrophic morphologies.

The PrP^res^ accumulation was also assessed by WB in all scrapie-infected mice, revealing a constant 19 kDa band pattern (Figure 5). The intensity of the PrP^res^ signal in tg338 mice was more homogeneous among individuals than in sheep, as expected for experimental infection in which the dose, age of inoculation, and time of sacrifice are controlled. No signal was detected in the control mice inoculated with normal brain homogenate, in agreement with the negative IHC results.

### 2.4. Inflammatory Effect of Scrapie Infection in CNS Alters Gene Expression of TLRs, MyD88, Trif, CD36, and Pro- and Anti-Inflammatory Cytokines

The upregulation of *TLRs* expression in response to PrP^Sc^ has been previously reported in experimental in vivo and in vitro conditions. Therefore, after characterizing the presence of the main prion-induced neuropathological changes in the CNS in the context of natural and experimental disease, we investigated whether these changes correlated with alterations in TLR gene expression. To this end, we performed qPCR using samples from the four brain regions of sheep to assess the expression of *MyD88* and *Trif*, the two main adaptor proteins implicated in TLR signaling; *CD36*, a scavenger receptor that cooperates with TLRs; and *TLR* mRNA. In addition, we measured the levels of four pro- and anti-inflammatory cytokines in the thalamus and hippocampus of sheep to further assess the inflammatory response. In tg338 mice, we measured the mRNA levels of *MyD88* and *TLR 1-9* in the thalamus. Genes for which significant differences were observed with respect to controls, and the corresponding fold changes, are shown in Appendix A, and the ΔCt values from the control and scrapie-infected sheep are represented in Appendix A.

The number of altered genes decreased in a caudo-rostral direction. The medulla oblongata of scrapie-infected sheep was the region in which the greatest changes in TLR gene expression were observed (Figure 6). In this region, the expression of *TLR2, 3, 4, 6, 7, 8, 9,* and *CD36* was significantly higher (*p <* 0.01) in the scrapie-infected versus control animals. In the thalamus, *TLR6, MyD88*, and *CD36* were significantly upregulated in the scrapie-infected group (*p <* 0.05), while *TLR2* and *TLR3* displayed a tendency towards upregulation (*p* < 0.1). In the frontal cortex (the rostral most area), *TLR4* and *CD36* were the only overexpressed genes (*p* < 0.05). Remarkably, contrasting expression patterns were found in the hippocampus of the scrapie-infected sheep, in which *TLR2* and *MyD88* were downregulated (*p <* 0.01) and *TLR1* showed a tendency towards downregulation (*p* < 0.1). In addition, the hippocampus was the only area in which *CD36* expression was not increased. Overall, *TLR4* was the gene that was most upregulated in scrapie-infected versus control sheep in all four brain regions, although no significant differences were observed in the thalamus or hippocampus due to intra-individual variability. The analysis of cytokine levels revealed the overexpression of *TGF-β*, *IL-10*, and *IL-6* in the thalamus of the scrapie-infected versus control sheep, but no alterations in the hippocampus (Figure 7).

To examine whether changes in TLR expression in the CNS in natural disease are comparable to those induced in the tg338 murine model after intracerebral infection with scrapie, the transcription levels of *TLRs* and *MyD88* were evaluated in the thalamus of tg338 mice (Figure 8). *TLR1* and *TLR2* mRNA levels were significantly increased (*p* < 0.01) compared with controls, and there was a tendency towards upregulation of *TLR7* (*p* < 0.1). The levels of *TLR8* expression were undetectable.

### 2.5. Assessment of TLR4 Protein Levels Confirms Upregulation Detected by qPCR in Scrapie-Infected Sheep

To confirm that the observed increase in *TLR4* mRNA translates to changes in protein levels, TLR4 was evaluated by WB in the medulla oblongata, thalamus, hippocampus, and frontal cortex of four scrapie-infected sheep and four control sheep (Figure 9). In agreement with the qPCR results, the TLR4 protein levels were significantly increased (*p* < 0.05) in all four brain regions of the scrapie-infected sheep.

## 3. Discussion

In recent years, the study of the neuroinflammatory response in prion diseases has led to a better understanding of the underlying mechanisms and the consequences. Neuroinflammation involves reactive microgliosis and astrogliosis [43], both of which can be detected before the onset of clinical signs and precede spongiosis and neuronal loss [7,8,18]. Whether microglial activation in prion diseases is beneficial or harmful remains a matter of debate. Some authors argue that microglial activation and proliferation contribute to the neuroinflammatory process and consequent neurodegeneration [8,18,44]. However, others point to an overall protective role of the microglia, and in vitro studies indicate that microglia can internalize and degrade PrP^Sc^ [27,45]. In line with these findings, in vivo experiments have shown that microglial depletion decreases the survival period, probably due to reduced PrP^Sc^ clearance [46,47]. Therefore, it is plausible that multiple reactive microglial phenotypes exist during the disease process, exerting a neuroprotective effect in early disease stages and, as the disease progresses, giving way to a proinflammatory phenotype that elicits detrimental effects [9,10]. One hypothesis proposes that TLR signaling mediates glial cell activation in prion diseases. The microglia uptake of amyloid fibers in Alzheimer’s disease is known to be promoted by TLR activation [48,49]. Thus, while the activation of TLRs on glial cells may play a role in the clearance of aggregated or abnormal proteins (e.g., prion protein), chronic activation may be detrimental to the host, resulting in neurotoxicity and neuronal cell death [50]. To date, data on the role of TLRs in prion pathogenesis are very limited. Few studies have measured changes in TLR expression in the CNS in murine models, and one has reported changes in TLR expression in the blood of lambs, with all cases using animals experimentally infected with prion disease [22,26,30].

We investigated the possible correlations between the neuropathological hallmarks of prion disease and TLR gene expression in four different brain regions of sheep naturally infected with scrapie. In all four brain areas, we observed significant differences in PrP^Sc^ deposition, spongiosis, astrogliosis, and microgliosis in the naturally infected versus control sheep. These differences were most evident in the medulla oblongata, the most caudal area. By contrast, the mildest lesions were observed in the frontal cortex, the most rostral area. This sequential pattern is in agreement with the route of prion neuroinvasion and dissemination throughout the CNS, beginning at entry sites in the spinal cord and obex and ultimately reaching the frontal cortex [1,5,51]. Unexpectedly, considering their anatomical proximity, we observed a drastic decrease in PrP^Sc^ deposition and vacuolization, and a significant decline in astrogliosis, when moving from the thalamus to the hippocampus, not following the gradual caudo–rostral progression of the pathology. A specific microglia morphology known as rod microglia was observed in the proximity of pyramidal cell neurons at CA3, but was absent from all of the other neuroanatomic regions studied. To our knowledge, this microglial profile has not been associated with neuroinflammation triggered by prions to date, although a recent study reported the presence of rod-shaped microglia in the cerebellum of human patients with CJD [52]. Although little studied since the first descriptions in the 1900s, rod microglia have been recently reported in neurological disorders such as epilepsy, Lewy body dementia, Huntington’s disease, and AD, specifically in moderately damaged areas of the cerebral cortex and hippocampus [40,41,53]. The phenotypic expression and functions of rod microglia are not yet clear. However, the fact that this morphological feature typically coincides with the presence of neuronal elements that are damaged or vulnerable to damage suggests a neuroprotective role [42,54,55]. Rod microglia are not thought to be associated with severe lesions, as the progression towards ameboid morphology is expected in these conditions [56]. The presence of rod microglia in the hippocampus of scrapie-infected sheep, but not in other brain regions, may be indicative of a neuroprotective environment, in good agreement with the limited neuronal loss observed in this area. Our analysis of TLR expression reveals a direct correlation between lesion severity and TLR gene overexpression that coincides with the aforementioned caudo–rostral progression of prion neuropathology. Conversely, TLR4 was similarly overexpressed in all areas, regardless of the lesion severity. TLR4 overexpression in areas of milder neuronal damage (i.e., the hippocampus and frontal cortex) may reflect a neuroprotective role, as previously described for phagocytic cells such as macrophages and microglia [23,57].

The upregulation of *TLR7* has been previously reported in mouse models of prion disease and in human patients [22,26,31]. We only observed the upregulation of *TLR7* in the medulla oblongata, the most damaged area in which the highest levels of spongiosis and PrP^Sc^ deposition were detected. Neuronal death via apoptosis has been previously associated with *TLR7* stimulation [58,59], and its overexpression in the medulla oblongata may be related to this mechanism [60]. Although the role of *TLR1* in neurodegenerative diseases is not yet clear, *TLR2* involvement in microglial activation has been increasingly demonstrated in amyotrophic lateral sclerosis, MS, and AD [61,62]. While the role of *TLR2* in prion diseases is not fully understood, beneficial effects have been proposed: survival time is reduced in mice that do not express *TLR2* following intracerebral inoculation with scrapie [22]. *TLR2* and *MyD88* overexpression may be also indicative of a proinflammatory microglia phenotype, which could explain the increase in *TNF-α* and *IL-6* that we observed in the thalamus [8,63]. Interestingly, experiments using EOC 13.31 cells, an immortalized microglia-like mouse cell line, have shown a dysregulation of the inflammatory response pathway in response to TLR2 activation, and suggested a link between TLR2 expression and the accumulation of microglia in a state not optimal for phagocytosis [26]. Therefore, *TLR2* overexpression in the medulla oblongata and the thalamus, the most damaged areas in which excessive levels of phagocytic microglia were also detected, suggests that the dysfunctional clearance of PrP^Sc^ may lead to sustained accumulation of PrP^Sc^ and neuronal damage [9,17].

Our findings in the hippocampus of scrapie-infected sheep reveal an opposite pattern compared with other analyzed brain areas, with the downregulation of *TLR1*, *TLR2*, and *MyD88* and no cytokine alterations. This may indicate that microglia respond differently in the hippocampus; indeed, it has been shown that TLR2 deficiency in primary microglial cell cultures, from neonatal mice (0–3 days old) and stimulated with neurotoxic peptide PrP106-126, shifts microglial activation from a neurotoxic to a neuroprotective phenotype [63]. However, the relevance of PrP106-126 peptide in prion pathology has been questioned [64]. CD36 is a different type of pattern recognition receptor, able to recognize endogenously derived ligands such as amyloid-forming peptides, that has established roles in the endocytic uptake of those components [65,66]. This receptor has been associated with a proinflammatory microglial status [67]. In vitro stimulation of BV-2 cells, a type of immortalized microglial cell, with PrP106-126 results in CD36 upregulation, increasing proinflammatory cytokines and iNOS and NO production [68,69]. Additionally, the recognition of β-amyloid peptide by CD36 triggers the assembly of a novel heterotrimeric complex CD36-TLR4-TLR6 that activates the innate immune response [70,71]. In our study, all regions showed significant upregulation of *CD36* except the hippocampus. The upregulation of *CD36*, *TLR4*, and *TLR6* in the most damaged areas, the medulla oblongata and obex, suggests the involvement of this triad in triggering a pro-inflammatory microglial status in response to prion infection. Conversely, the absence of *CD36* and *TLR6* overexpression in the hippocampus suggests that this heterotrimer is not formed, indicative again of a neuroprotective environment in this brain region.

It remains to be explained why microglia respond differently in this brain region. While prion strain-specific cell tropism could determine the pattern of microgliosis and astrogliosis, recent findings suggest that both reactions are mainly influenced by the brain region [18,72]. In fact, neither microglia nor astroglia respond uniformly across the CNS, and this region-specific response could result in the selective vulnerability of some brain regions in prion diseases [16,18]. In this regard, it has been postulated that the inflammatory response of microglia to prion infection is regulated by the sialylation of PrP^Sc^ [14,73,74,75], and that PrP^Sc^ sialylation is brain region-dependent [18]. Specifically, a higher level of sialylation of PrP^Sc^ is found in the hippocampus and cortex than the thalamus and brainstem, suggesting a potential role in the selective vulnerability of these brain regions [18,73,75]. The high level of PrP^Sc^ sialylation in the hippocampus may be associated with decelerated prion replication, leading to the distinctive prion-induced microglial activation in this region, consistent with the reduced susceptibility of this region implied by our findings.

Interestingly, previous findings suggest that the hippocampus may be protected from prion neurotoxicity in natural CJD infection [76,77], and detailed neuropathological studies of CJD cases have reported milder lesions in the hippocampus than in other brain regions [76]. Specifically, the archicortex, which primarily comprises the hippocampus, appears to be relatively spared compared with other cortical regions in CJD [77]. This mild hippocampal involvement described in natural CJD is in agreement with the present findings in sheep naturally infected with scrapie. It is interesting that this partial protection against at least two natural prion diseases occurs in the hippocampus, which is phylogenetically the oldest region of the cerebral cortex and consists of the most basic type of cortical tissue. Further studies will be required to explore the relevance of this correlation.

In summary, our results reveal a particularly mild neuropathology in the hippocampus of natural scrapie-infected sheep, characterized by lower levels of spongiosis, PrP^Sc^ deposition, and astrogliosis than expected given the caudal-to-rostral spread of scrapie lesions. Moreover, the presence in the hippocampus of an exclusive microglial morphology, rod microglia, which may play a neuroprotective role [54], together with a distinct pattern recognition receptor (TLR genes and *CD36*) gene expression profile, further differentiate this brain region from the other neuroanatomic regions in natural scrapie-infected sheep. These findings suggest a degree of neuroprotection against natural prion infection in the hippocampus that merits further investigation.

Neurodegenerative diseases caused by protein misfolding involve a vicious cycle of inflammation consisting of misfolded protein accumulation, glial activation, and the release of glial inflammatory mediators, which exacerbate protein deposition and neuroinflammation. Interrupting this vicious cycle by targeting microglia activation using specific TLR inhibitors at a specific disease stage may constitute a promising approach to limit further neuroinflammation. Our findings highlight TLR2 downregulation as a potential target for such an approach, as this may induce a shift in microglia from a neurotoxic to a neuroprotective phenotype [63].

Finally, while great progress has been made in characterizing the diversity of glial phenotypes using mouse models of neurodegenerative diseases, whether mouse models faithfully reflect key aspects of prion diseases is a matter of some debate [78,79]. Our results in the tg338 transgenic model reproduced the common pathological signs of prion infection, including marked PrP^Sc^ deposition, neuropil spongiosis, astrogliosis, and microgliosis. However, infected mice displayed significant overexpression of *TLR1* and *TLR2* and a tendency towards *TLR7* overexpression. While the upregulation of these genes has been previously described in other mouse models infected with scrapie [22,26], this pattern contrasts with our findings in the ovine brain. Remarkably, in the mouse brain we observed no alterations in the expression of *TLR4*, the gene for which the greatest changes in expression were observed in the sheep samples. These conflicting findings may be a consequence of the different route of infection or/and the prion protein expression levels in tg338 mice [80,81]. Nonetheless, our results ultimately indicate that the intracerebral inoculation of scrapie in ovinized tg338 mice does not reproduce the immune response observed in natural scrapie infection.

## 4. Materials and Methods

### 4.1. Scrapie-Infected and Control Sheep

Twenty-one female *Rasa Aragonesa* sheep (aged from 2–6 years) were included in the present study. All were genotyped for *PRNP* polymorphisms, as previously reported [82], and were found to display an ARQ/ARQ genotype. Control animals (*n* = 8) were selected from a flock in which no scrapie cases had been reported. Scrapie-infected animals (*n* = 13) were obtained from scrapie-affected flocks and had been diagnosed by immunohistochemistry (IHC) of rectal mucosa biopsies. Infection was confirmed by *post-mortem* immunodetection of PrP^Sc^ in the obex following published criteria [83]. The animals were euthanized by an intravenous overdose of barbiturate and exsanguination. At the time of euthanasia, all scrapie-infected sheep showed clinical signs of diverse severity: some animals displayed incipient signs such as pruritus of the back and flanks after digital stimulation and mild reduction of the body condition, whereas others displayed advanced clinical signs such as spontaneous scratching of the tail root, lumbar area, and limbs, neurological signs including ataxia and head tremors, and teeth grinding, wool loss, and intense weight loss [84].

### 4.2. Infection of Tg338 Mouse

To evaluate *TLR* geneexpressions in the brain of a murine model of scrapie, ten-week-old tg338 mice (*n* = 8) (overexpressing the ovine VRQ/VRQ PrP^C^ 8- to 10-fold [85]) were inoculated with a brain pool from a natural scrapie-infected *Rasa Aragonesa* sheep sacrificed at the clinical stage. The mice were inoculated with 20 uL of the scrapie inoculum (diluted 2% *w*/*v* in PBS) into the right cerebral hemisphere under isoflurane anesthesia. Intracerebral injections were performed using a 50 μL syringe and a 25G needle. After inoculation, the mice were administered a subcutaneous injection of buprenorphine (0.3 mg/kg) to induce analgesia. As controls, tg338 mice (*n* = 8) were inoculated with brain homogenate from a scrapie-negative sheep following the same procedure described above. The mice were monitored for the development of clinical signs and euthanized by cervical dislocation when terminal signs of disease such as severe ataxia and inability to feed appeared. Mice infected with the scrapie-positive inoculum displayed a mean survival time of 187 ± 26 dpi.

### 4.3. Tissue Collection

Samples from the CNS were collected and divided sagittally into two halves; one was fixed in 10% neutral-buffered formalin for histopathological and immunohistochemical analysis, and the other was directly frozen and maintained at −80 °C for protein analysis or stabilized in RNAlater^TM^ Solution (Invitrogen^TM^, Waltham, MA, USA) for RNA extraction and then frozen and stored at −80 °C.

### 4.4. PRNP Sequencing

DNA was extracted from blood samples with Speedtools Tissue DNA Extraction kit (Biotools, Madrid, Spain) according to the manufacturer’s instructions. PCR amplification and sequencing were done as described previously [82].

### 4.5. Immunohistochemistry

Formalin-fixed tissues were processed according to standard histopathological procedures. Tissue sections were paraffin-embedded, cut into 4 µm thick sections, and stained with hematoxylin–eosin (HE) for the evaluation of vacuolation and neuropil spongiosis. IHC for PrP^Sc^ detection was performed using the mouse monoclonal primary antibody L42 in sheep (1:500 dilution at room temperature for 30 min) (R-Biopharm, Darmstadt, Germany) and rabbit polyclonal antibody R486 in mice (1:8000 dilution, overnight at 4 °C) (R. Jackman, unpublished) as previously described [86,87]. Sections were also subjected to conventional immunostaining for the astrocyte marker glial fibrillary acidic protein (GFAP) (1:500; Dako, Glostrup, Denmark), and the microglia marker ionized calcium-binding adaptor molecule 1 (Iba1) (1:1000; Wako, Richmond, VA, USA), according to published protocols [88].

All histological and IHC evaluations were performed by two veterinary pathologists blinded to the clinical data. Assessments of spongiosis and PrP^Sc^ staining intensity were semi-quantitatively performed and adapted following the criteria described in previous studies: vacuolation of the neuropil and the perikarya was scored from 0 (absent) to 5 (very numerous and confluent) [51], the PrP^Sc^ signal was quantified based on the extent of immunostaining from 0 (no labelling) to 5 (intense uniform labeling) as previously reported [89], and the extent of GFAP and Iba1 immunolabelling was scored on a scale ranging from 0 to 5 (0 = weak staining; 5 = substantial immunolabelling throughout the region) as described [88]. Four brain regions were evaluated: frontal cortex (Fc) thalamus (Th), hippocampal formation (Hc), and medulla oblongata (Mo), and each area was globally analyzed for the scoring and graphically represented as mean ± standard error.

### 4.6. Western Blot

100 mg of brain tissue of each brain area (Fc, Th, Hc, and Mo) from 13 scrapie-infected sheep, 8 with advanced clinical signs and 5 with incipient clinical signs, were homogenized in 1 mL of lysis buffer. Hemiencephalons of the 8 infected and 8 control mice were homogenized at 10% (*w*/*v*) in lysis buffer. Tissue samples were homogenized in grinding tubes (Bio-Rad, Hercules, CA, USA) using a TeSeEPrecess 48 TM homogenizer (Bio-Rad, Hercules, CA, USA) and the protein concentration was measured using the Pierce^TM^ BCA Protein Assay kit (ThermoScientific^TM^, Waltham, MA, USA) according to the manufacturer’s instructions.

For the PrP^res^ analysis, equal protein amounts from tissue homogenates were incubated for 10 min at 37 °C with proteinase K solution, as previously described [90]. The resulting samples were subjected to electrophoresis in 12% Criterion^TM^ XT Bis-Tris Protein Gel (Bio-Rad, Hercules, CA, USA) and transferred to PVDF membranes that were blocked for 1 h with 2% non-fat dry milk in TBST (Tris-buffered saline with 0.1% Tween 20). For immunoblotting, the membranes were incubated overnight at 4 °C with Sha31 primary antibody (SPI-Bio, Montigny-le-Bretonneux, France) at a concentration of 1µg/mL followed by 1 h incubation at room temperature (RT) with horseradish peroxidase conjugated anti-mouse IgG secondary antibody (1:5000) (Santa Cruz Biotechnology, Dallas, TX, USA). Immunoreactivity was detected using the chemiluminescent substrate Immobilon Crescendo Western HRP (Merck, Darmstadt, Germany).

The TLR4 protein expression was analyzed from the tissue homogenates mixed with 2×Laemmli Sample buffer (Bio-Rad, Hercules, CA, USA) according to the manufacturer’s instructions. Forty micrograms of total protein were loaded per well, run in 7.5% Criterion^TM^ TGX^TM^ Precast Midi Protein Gel (Bio-Rad, Hercules, CA, USA) and transferred to PVDF membranes, which were then blocked for 2 h with 4% bovine serum albumin (BSA) (Merck, Darmstadt, Germany) in TBST at RT. The membranes were incubated overnight at 4 °C with rabbit polyclonal anti-TLR4 antibody (Novus Biological, Minneapolis, MN, USA) at a concentration of 0.5 µg/mL, and then washed and incubated with goat anti-rabbit IgG (H + L) HRP secondary antibody (ThermoScientific^TM^, Waltham, MA, USA) at 1:20,000 for 1 h at RT. Blots were visualized as described above. Next, the membranes were stripped with Restore^TM^ Western Blot Stripping buffer (ThermoScientific^TM^, Waltham, MA, USA) for 15 min at 37 °C, washed, and blocked again. Then, the membranes were incubated overnight at 4 °C with mouse monoclonal β-actin primary antibody (Santa Cruz Biotechnology, Dallas, TX, USA) at 1:1000, washed, and incubated with anti-mouse m-IgGк BP-HRP secondary antibody (Santa Cruz Biotechnology, Dallas, TX, USA) for 1 h. After washing, the blots were developed as described above.

Densitometries were carried out with ImageJ software and the values were normalized using β-actin. The normalized values were represented with GraphPad Prism 6.0 (San Diego, CA, USA). The statistical analyses to compare the infected and control groups were performed with a Student’s *t*-test, and the equality of variances was determined by Levene’s test using the SPSS software (SPSS Statistics for Windows, Version 17.0, Chicago, IL, USA). Differences between groups were considered statistically significant at * *p* <0.05.

### 4.7. RNA Extraction, cDNA Synthesis, and Gene Expression

Sheep total RNA was extracted from 90 mg of tissue samples from the frontal cortex, thalamus, hippocampus, and medulla oblongata. Mouse brains were divided at the midline and total RNA was extracted from <90 mg obtained from the thalamic area. Tissues were homogenized using a TeSeEPrecess 48^TM^ homogenizer (Bio-Rad) with RNeasy Lipid Tissue Mini Kit (Qiagen, Hilden, Germany) combined with TURBO DNase (Invitrogen^TM^, Waltham, MA, USA) to remove possible genomic DNA contamination. RNA concentration was determined spectrophotometrically with a NanoDrop spectrophotometer (Thermo Fisher Scientific, Waltham, MA, USA), and for each sample, 260/280 and 260/230 ratios were analyzed to verify the sample purity. One microgram of complementary DNA (cDNA) was synthesized using qScript^TM^ cDNA SuperMix (Quantabio Biosciences^TM^, Beverly, MA, USA) according to the manufacturer’s instructions. In addition, the effectiveness of the DNase treatment was assessed in RT-negative samples. After reverse transcription, the same batch of diluted cDNA was subjected to qPCR to amplify *TLRs*.

Two commonly used housekeeping (HK) genes were selected to normalize the expression of the targets genes: glyceraldehyde-3-phosphate dehydrogenase (*GAPDH*) and actin-beta (*ACTβ*) [91]. The stability of this HK gene was verified under our experimental conditions. The messenger RNA (mRNA) expression was determined by qPCR for 1 to 10 ovine *TLR* genes, 1 to 9 murine *TLR* genes, and the *MyD88* gene for both species. Two proinflammatory (*TNF-α* and *IL-6*) and two anti-inflammatory (*IL-10* and *TGF-β*) cytokines were also studied in the sheep thalamus and hippocampus. Primer sequences and efficiencies had been previously published or were designed using the Primer3Plus tool [92] (Table 2). We verified the efficiency of each gene generating a standard curve by amplifying 1:2 serial dilutions of control cDNA, and then checking for linearity between the initial template concentration and cycle threshold (Ct) values. All genes showed a correlation coefficient between 0.9 and 0.99, with a slope value of the standard curves in the ranges of −3.2 to −3.5 and qPCR efficiency of 90–110%.

The qPCR reactions were run using Applied Biosystems^TM^ QuantStudio^TM^ 5 Real-Time PCR System, 96-well with universal amplification conditions: an initial activation and cDNA denaturation step of 10 min at 95 °C, followed by 40 cycles of 3 s at 95 °C and 30 s at 60 °C. To identify the presence of nonspecific PCR amplicons or high levels of primer dimers, we performed a dissociation curve protocol after each qPCR reaction. Each sample was analyzed in triplicate in a total reaction volume of 10 µL, consisting of 15 ng of cDNA, 5 µL Fast SYBR Green Master Mix (2X) (ThermoFisher Scientific, Waltham, MA, USA), and the required amount of forward and reverse primers (Table 2). Nuclease-free water was added to a final volume of 10 µL. The levels of gene expression were determined using the comparative Ct method. The results were represented as fold-change and the gene expression differences relative to the mean level of the control group scaled to 1.

### 4.8. Data Analysis and Statistics

All quantitative data collected were tested for normality with the Shapiro–Wilk W test. Histological and immunohistochemical differences between the infected and control groups were evaluated using a Student’s *t*-test or Mann–Whitney U test depending on the parametric or non parametric data distribution. Statistical differences between the four different brain regions in scrapie infected-sheep were determined using one-way analysis of variance (ANOVA) followed by a Bonferroni post hoc test or Kruskal–Wallis test, depending on the parametric or non parametric data distribution. Statistical analyses of the qPCR data were conducted from the mean ΔCt values for each gene. For the statistical comparison of infected and control groups, a Student’s *t*-test or Mann–Whitney U test were performed depending on the normal distribution of each gene, and the equality of variances was determined by a Levene’s test. Differences in expression were considered to be significant at *p* < 0.05. The following notations were used to denote *p*-values in the figures: * *p* < 0.05; ** *p* ≤ 0.01; # *p* < 0.1. SPSS (SPSS Statistics for Windows, Version 17.0, Chicago, IL, USA) software was used for the statistical analyses. Graphs were generated with GraphPad Prism 6.0 (San Diego, CA, USA) and the data shown in the figures represent the mean and the standard error of the mean (mean ± SEM).

## 5. Conclusions

To our knowledge, the present study is the first describing the expression levels of TLR genes in different brain regions of natural scrapie-infected sheep and ovinized tg338 mice experimentally infected with scrapie. Our study clearly shows that TLRs, and especially *TLR4* in sheep and *TLR1* and *TLR2* in mice, are involved in the pathogenesis of scrapie. In addition, in contrast to all other regions studied, the distinctive profile of TLR gene expression, together with the unique microglial morphology and mild neuropathology observed in the hippocampus, suggests a brain region-specific immune response in natural scrapie infection. However, further studies will be necessary to characterize *TLR* expression in microglia, astroglia, and neurons in scrapie infection in order to understand the precise contribution of each cell type to neuroinflammation. Furthermore, it is yet to be determined whether TLR activation is a direct response to prion toxicity or occurs secondary to other inflammatory mechanisms. TLRs constitute a promising target for therapeutic approaches to prion diseases, and therefore a better understanding of TLR-regulated neuroinflammatory responses will be needed to ensure further advances in this area.

## Figures and Tables

**Figure 1 ijms-23-03579-f001:**
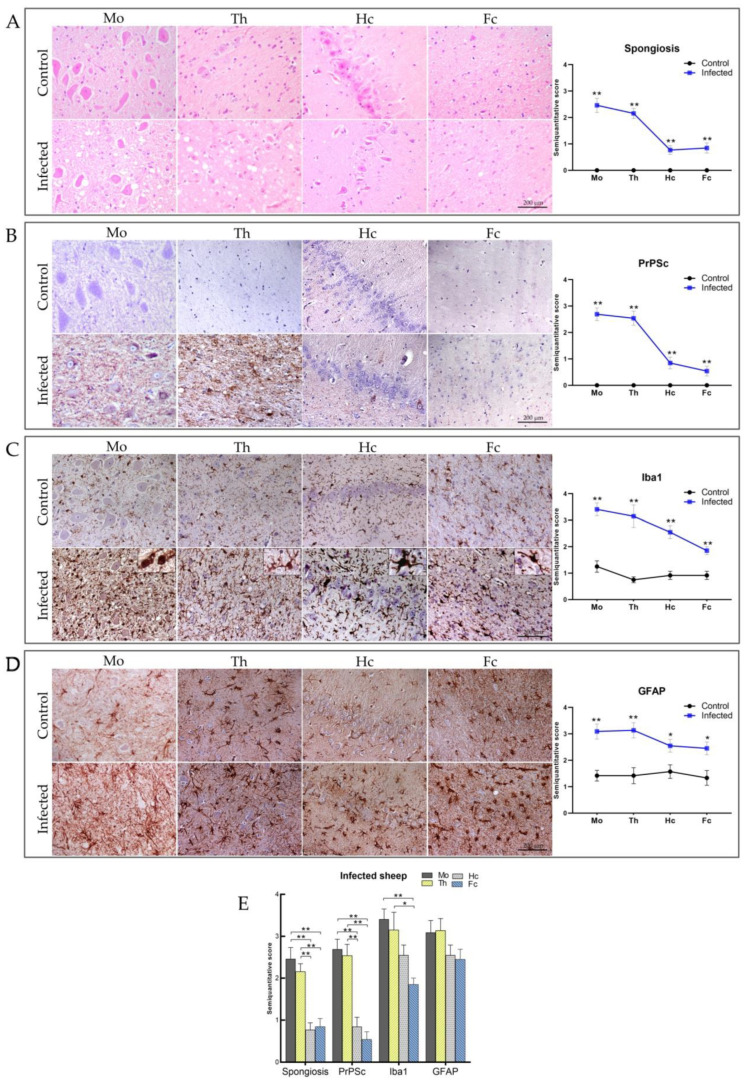
Representative neuropathology and immunohistochemical assessment of spongiosis, PrP^Sc^ deposition, microgliosis, and astrogliosis in the CNS of control and scrapie-infected sheep in the medulla oblongata (Mo), thalamus (Th), hippocampus (Hc), and frontal cortex (Fc). Graphs depict semi-quantitative assessment values for each parameter analyzed. (**A**) Hematoxylin–eosin staining showing spongiosis changes in the CNS from scrapie-infected sheep, and an absence of spongiosis in control sheep. (**B**) Detection of PrP^Sc^ with L42 antibody in scrapie-infected sheep brain regions. No immunolabelling was detected in the control group. (**C**) Ionized calcium-binding adaptor molecule-1 (Iba1) immunostaining showing increased microglial reactivity in scrapie-infected versus control sheep. (**D**) Glial fibrillary acidic protein (GFAP) immunostaining displaying increased reactive astrogliosis in scrapie-infected sheep compared to control sheep. (**E**) Statistical comparisons of spongiosis, PrP^Sc^ deposition, Iba1, and GFAP intensity between brain regions. Scores range from 0 (negative) to 5 (maximum intensity). Significant differences in spongiosis, PrP^Sc^, Iba1, and GFAP were determined using Student’s *t*-test or Mann–Whitney U test for normally and non-normally distributed data, respectively. Statistical differences between brain areas were determined using one-way analysis of variance (ANOVA) or Kruskal–Wallis test for normally and non-normally distributed data, respectively. * *p* < 0.05, ** *p* < 0.01. Scale bar = 200 µm.

**Figure 2 ijms-23-03579-f002:**
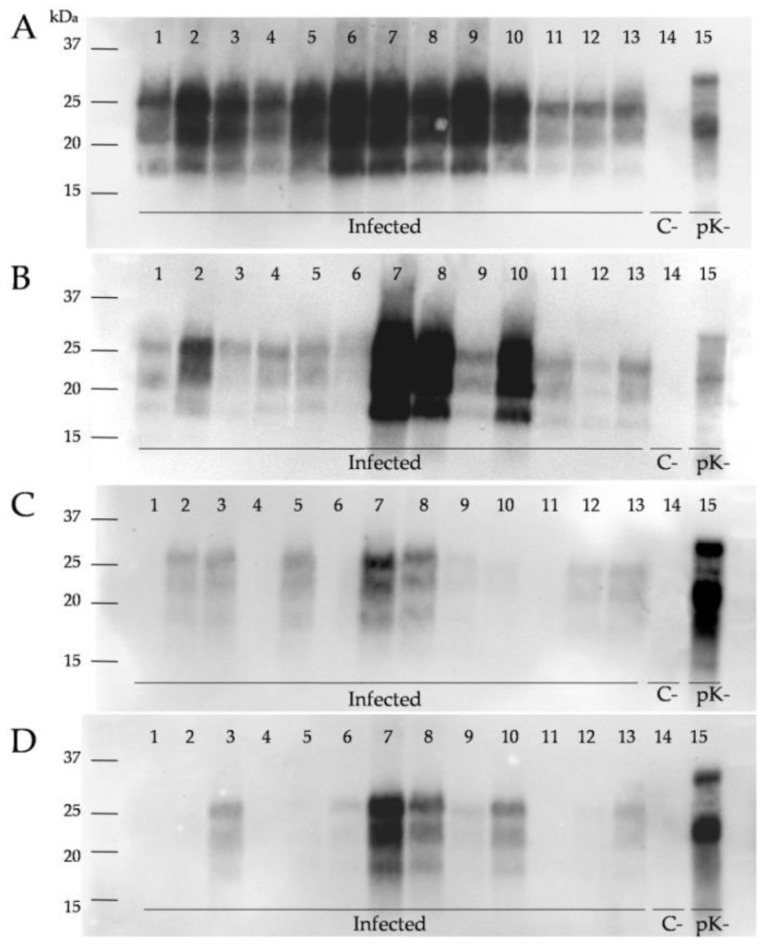
Western blot detection of PrP^res^ in the CNS of control and natural scrapie-infected sheep. Equal protein quantities were loaded for adequate comparison, and immunoblots were run using SHA31 monoclonal antibody for analysis of samples from the (**A**) medulla oblongata, (**B**) thalamus, (**C**) hippocampus, and (**D**) frontal cortex from scrapie-infected sheep with incipient clinical signs (samples 1–5) and with advanced clinical signs (samples 6–13). Line 14 shows the negative control and line 15 a sample of non-infected sheep brain homogenate not subjected to pK digestion.

**Figure 3 ijms-23-03579-f003:**
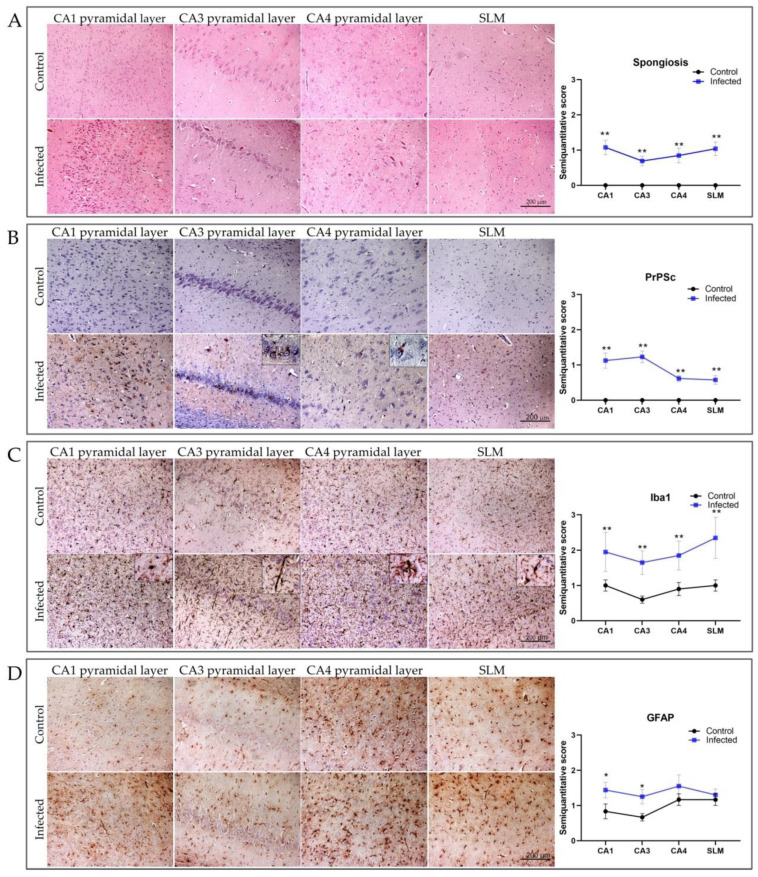
Distribution of neuropathological lesions (HE), PrP^Sc^ deposition, astrogliosis (GFAP), and microgliosis (Iba1), in four different hippocampal regions in scrapie-infected and control sheep: pyramidal cell layers *CornuAmmonis* (CA) 1, 3, and 4, and the *stratum lacunosum-moleculare* (SLM). Graphs depict semi-quantitative assessment values for each parameter analyzed. (**A**) Mild spongiform changes were observed in all hippocampal regions analyzed. (**B**) Intraneuronal PrP^Sc^ pattern (L42) in CA1, CA3, and CA4 and widespread fine granular PrP^Sc^ pattern in the SLM. (**C**) Increased Iba1 staining in scrapie-infected versus control sheep. Magnified images show some of the morphological differences observed. (**D**) Signs of moderate reactive astrogliosis in scrapie-infected sheep compared with controls. Scores range from 0 (negative) to 5 (maximum intensity). Significant differences were determined using Student’s *t*-test or Mann–Whitney U test for normally and non-normally distributed data, respectively. * *p* < 0.05, *** p* < 0.01. Scale bar = 200 µm.

**Figure 4 ijms-23-03579-f004:**
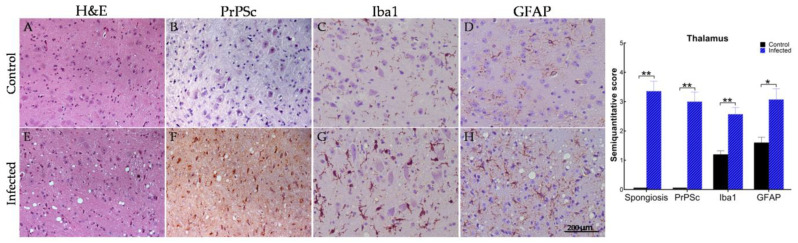
Thalamus sections from scrapie-infected (intracerebral inoculation) and age-matched tg338 control mice immunostained to assess spongiosis (hematoxylin–eosin), PrP^Sc^ deposition (R486 antibody), astrocytes (glial fibrillary acidic protein; GFAP), and microglia (ionized calcium-binding adaptor molecule-1; Iba1). (**A**,**E**) Severe spongiform lesions located mainly in the neuropil of scrapie-infected versus control mice. (**B**,**F**) Several PrP^Sc^-positive neurons are evident in scrapie-infected mice, but absent from controls. (**C**,**G**) Activated microglia displaying more intense immunostaining and thick processes in scrapie-infected mice versus controls. (**D**,**H**) Increased astrogliosis with hypertrophic cell bodies and processes in scrapie-infected mice versus controls. Scores range from 0 (negative) to 5 (maximum intensity). Significant differences were determined using Student’s *t*-test or Mann–Whitney U test for normally and non-normally distributed data, respectively. * *p* < 0.05, ** *p* < 0.01. Scale bar = 200 µm.

**Figure 5 ijms-23-03579-f005:**
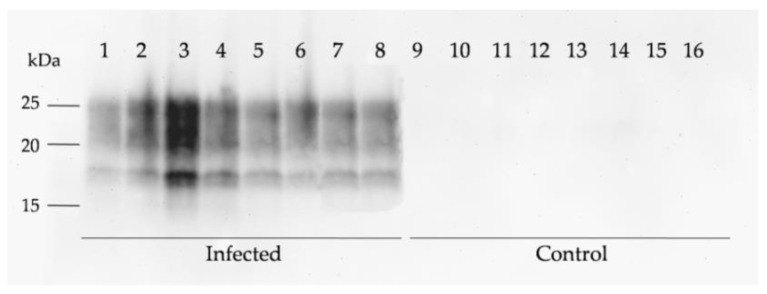
Western blot detection of PrP^res^ accumulation in the CNS of clinically affected tg338 mice intracerebrally inoculated with scrapie compared with matched controls. Equivalent protein quantities were loaded for adequate comparison and immunoblots were run using SHA31 monoclonal antibody. Molecular-weight markers (kDa) are indicated on the left side of the immunoblot.

**Figure 6 ijms-23-03579-f006:**
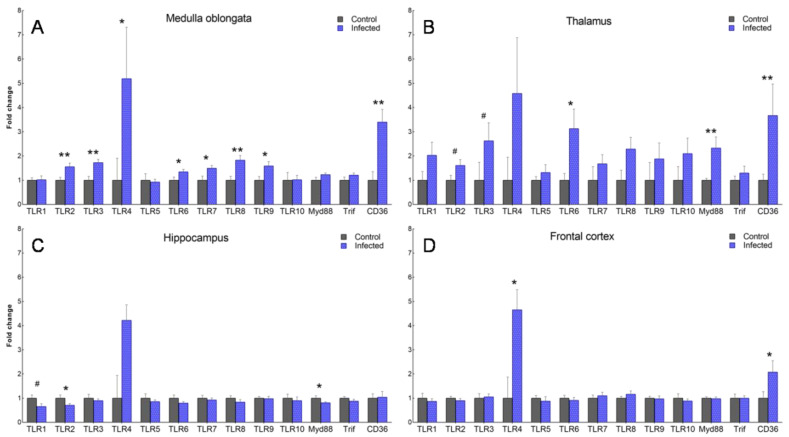
Gene expression of *TLR 1-10*, *MyD88, Trif,* and *CD36* in the medulla oblongata (**A**), thalamus (**B**), frontal cortex (**C**), and hippocampus (**D**) of control and scrapie-infected sheep. Data are represented as the mean fold ± SEM increase with respect to control sheep (expressed as a score of 1). Mean scores were compared using the Student’s *t*-test or Mann–Whitney U test for normally and non-normally distributed data, respectively. # *p* < 0.1, * *p* < 0.05, ** *p* < 0.01.

**Figure 7 ijms-23-03579-f007:**
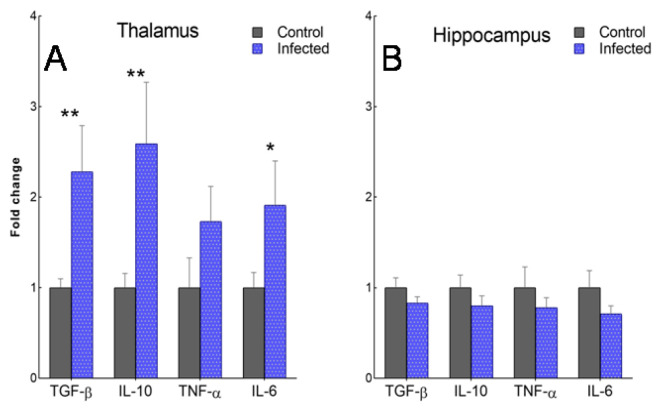
Gene expression of the anti-inflammatory cytokines *TGF-β* and *IL-10*, and the proinflammatory cytokines *TNF-α* and *IL-6,* in the thalamus (**A**) and hippocampus (**B**) of control and naturally scrapie-infected sheep. Data are represented as the mean ± SEM fold change with respect to controls (expressed as a score of 1). Mean scores were compared using the Student’s *t*-test or Mann–Whitney U test for normally and non-normally distributed data, respectively. ** p* < 0.05, ** *p* < 0.01.

**Figure 8 ijms-23-03579-f008:**
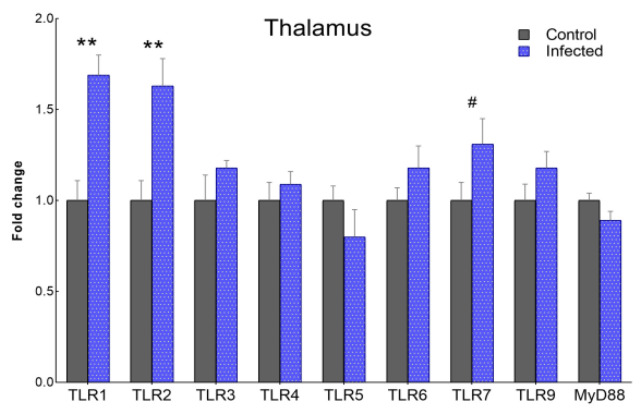
Gene expression of *TLR1-7*, *TLR9,* and *MyD88* in the thalamus of clinically affected tg338 mice intracerebrally inoculated with scrapie-positive inoculum versus controls (scrapie-negative inoculum). Results are expressed as the mean ± SEM fold change with respect to controls (expressed as a score of 1). Mean scores were compared using the Student’s *t*-test or Mann–Whitney U test for normally and non-normally distributed data, respectively. *# p* < 0.1, *** p* < 0.01.

**Figure 9 ijms-23-03579-f009:**
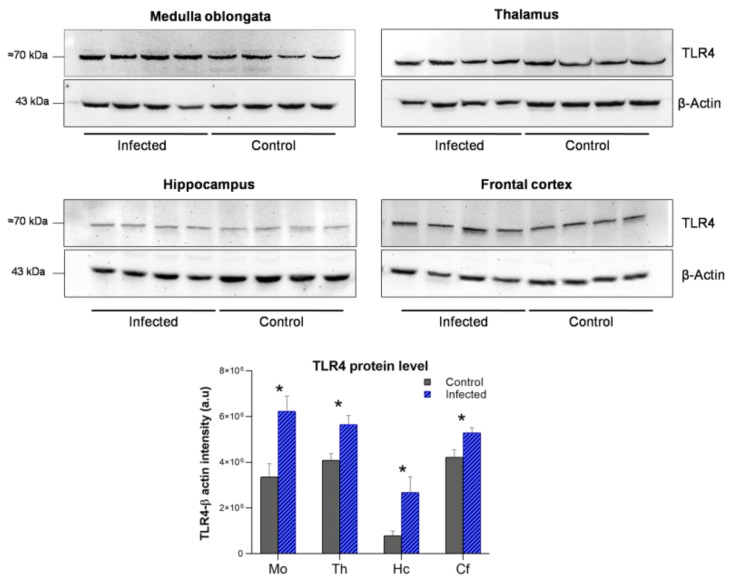
TLR4 protein expression in the medulla oblongata (Mo), thalamus (Th), hippocampus (Hc), and frontal cortex (Fc) of scrapie-infected sheep versus control sheep (*n* = 4 per group). β-actin was used as a loading control. The graph depicts densitometry values and indicates a significant increase in TLR4 expression in all regions in scrapie-infected versus control sheep. Data are expressed as the mean ± SEM. Mean scores were compared using the Student’s *t*-test. * *p* < 0.05.

**Table 1 ijms-23-03579-t001:** PrP^Sc^ detection by immunohistochemistry (IHC) and Western blot (WB) in the medulla oblongata (Mo), thalamus (Th), hippocampus (Hc), and frontal cortex (Fc) of scrapie-infected sheep.

		Mo		Th		Hc		Fc	
Clinical Stage	ID	IHC	WB	IHC	WB	IHC	WB	IHC	WB
Incipient	1	+	+	+	+	+	−	−	−
Incipient	2	+	+	+	+	+	+	−	−
Incipient	3	+	+	+	+	+	+	+	+
Incipient	4	+	+	+	+	+	−	−	−
Incipient	5	+	+	+	+	+	+	+	+
Advanced	6	+	+	+	+	+	−	+	+
Advanced	7	+	+	+	+	+	+	+	+
Advanced	8	+	+	+	+	+	+	+	+
Advanced	9	+	+	+	+	+	+	+	+
Advanced	10	+	+	+	+	+	+	+	+
Advanced	11	+	+	+	+	+	−	+	−
Advanced	12	+	+	+	+	+	+	+	+
Advanced	13	+	+	+	+	+	+	+	+

**Table 2 ijms-23-03579-t002:** Primers used for quantification of *gene expressions*.

Gene	Forward (F) and Reverse (R) Primer Sequence (5′-3′)	Primer Concentrations innMUsed for qPCR	Accession Number	Reference
Ovine Primers				
*TLR1*	F CCCACAGGAAAGAAATTCCA R GGAGGATCGTGATGAAGGAA	900	NM_001135060.2	[93]
*TLR2*	F ACGACGCCTTTGTGTCCTAC R CCGAAAGCACAAAGATGGTT	900	NM_001048231.1	[93]
*TLR3*	F GAGGCAGGTGTCCTTGAACT R GCTGAATTTCTGGACCCAAG	900	NM_001135928.1	[93]
*TLR4*	F ACTGACGGGAAACCCTATCC R CAGGTTGGGAAGGTCAGAAA	900	NM_001135930.1	[93]
*TLR5*	F AAAACCACATCGCCAACATC R CATCAGATGGAACTGGGACA	900	NM_001135926.1	[93]
*TLR6*	F CAAAGCAGGGAACAATCCAT R CCACAATGGTGACAATCAGC	900	NM_001135927.1	[93]
*TLR7*	F ACTCCTTGGGGCTAGATGGT R GCTGGAGAGATGCCTGCTAT	900	NM_001135059.1	[93]
*TLR8*	F CACATCCCAGACTTTCTACGA R GGTCCCAATCCCTTTCCTCTA	900	NM_001135929.1	[93]
*TLR9*	F CTCGTATCCCTGTCGCTGAG R CACCTCCGTGAGGTTGTTGT	900	NM_001011555.1	[93]
*TLR10*	F TCTGCCTGGGTGAAGTATGA R AATGGCACCATTCAGTCTGG	900	NM_001135925.1	[93]
*TNF-α*	F CAAATAACAAGCCGGTAGCC R TGGTTGTCTTTCAGCTCCAC	200	NM_001024860.1	[94]
*TGF-β*	F TTGACGTCACTGGAGTTGTG R CGTTGATGTCCACTTGAAGC	200	NM_001009400.2	[94]
*IL-10*	F TTAAGGGTTACCTGGGTTGC R TTCACGTGCTCCTTGATGTC	200	NM_001009327.1	[94]
*IL-6*	F CGAGTTTGAGGGAAATCAGG R GTCAGTGTGTGTGGCTGGAG	300	NM_001009392.1	[95]
*MyD88*	F CTGCAAAGCAAGGAATGTGA R AGGATGCTGGGGAACTCTTT	400 300	NM_001166183.1	Designed
*Trif*	F GCACGTCTAGCCTGCTTACC R AGGTGTTGGTCACCTTCCTG	300	XM_042250120.1	Designed
*CD36*	F GCAAAACGGCTGCAGATCAA R AGCAATGGTGGCAGTCTCAT	300	XM_012176565.4	Designed
*ACTβ*	F CTGGACTTCGAGCAGGAGAT R GATGTCGACGTCACACTTC	600	NM_001009784	[94]
*GAPDH*	F TCCGGGAAGCTGTGGCGTGA R GGGATGACCTTGCCCACGGC	500	NM 001190390.1	[96]
Murine Primers				
*TLR1*	F CTGGACCCAGAGTTTGTTAGTTTT R GCTCATTATCCTGTTGTTGTGAAG	150 300	XM_006503856.3	Designed
*TLR2*	F GCCACCATTTCCACGGACT R GGCTTCCTCTTGGCCTGG	500	NM_011905	[97]
*TLR3*	F TTGTCTTCTGCACGAACCTG R CCCGTTCCCAACTTTGTAGA	300	XM_006509283.4	Designed
*TLR4*	F AGAAATTCCTGCAGTGGGTCA R CTCTACAGGTGTTGCACATGTCA	500	NM_021297	[97]
*TLR5*	F GCCACATCATTTCCACTCCT R ACAGCCGAAGTTCCAAGAGA	200	XM_017321704.2	[98]
*TLR6*	F ACACAATCGGTTGCAAAACA R GGAAAGTCAGCTTCGTCAGG	300 400	NM_001384171.1	Designed
*TLR7*	F GGTATGCCGCCAAATCTAAA R GCTGAGGTCCAAAATTTCCA	400 500	XM_006528713.2	Designed
*TLR8*	F GAAGCATTTCGAGCATCTCC R GAAGACGATTTCGCCAAGAG	200	XM_017318405.2	[98]
*TLR9*	F CAACCTCAGCCACAACATTC R CACACTTCACACCATTAGCC	200	NM_031178.2	[98]
*MyD88*	F CATGGTGGTGGTTGTTTCTGAC R TGGAGACAGGCTGAGTGCAA	500	NM_010851.3	[99]
*GAPDH*	F CCTCGTCCCGTAGACAAAATG R TGAAGGGGTCGTTGATGGC	250	XM_036165840.1	[100]
*ACTβ*	F CTTCTTTGCAGCTCCTTCGTT R TTCTGACCCATTCCCACCA	250	NM_007393.5	[100]

## Data Availability

The data presented in this study are available within the article text, figures and Appendix A.

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
