# Peer review of "Distinctive Toll-like Receptors Gene Expression and Glial Response in Different Brain Regions of Natural Scrapie"

_ijms, 2022, doi:10.3390/ijms23073579_

Round 1

Reviewer 1 Report

The manuscript by García-Martínez et al. is improved over the original submission. I have only a few concerns that require clarification, specifically in the Discussion section.

Lines 432-444 “Moreover, previous studies have linked TLR2 expression with the accumulation of microglia in a state not optimal for phagocytosis [26]”

This statement needs serious clarification. These studies were not performed with microglia but were completed with a mouse cell line called EOC 13.31. Like BV-2 cells, these immortalized cells have various characteristics that are microglia-like and macrophage-like but have a very different transcriptional signature. We need to be careful not to misrepresent these findings from immortalized cells grown in monoculture to cells found in the CNS.

Lines 440-442 “This may indicate that microglia respond differently in the hippocampus; indeed, it has been shown that TLR2 deficiency shifts prion-induced microglial activation from a neurotoxic to a neuroprotective phenotype [63].”

Again, this statement requires clarification. This study was performed with culture mouse microglia from neonates (which do not act the same as adult microglia when stimulated). Furthermore, this was not prion-induced, rather these studies were completed using the PrP106-126 fragment, that has been found to be irrelevant during actual prion infection. Again, I feel care is needed to correctly refer to these studies. The authors are comparing their hippocampal findings to in vitro results using neonates stimulated with a peptide fragment whose relevance to disease is questionable.

Lines 446-448 “In vitro stimulation of microglia with PrP106-126 results in CD36 upregulation, increasing proinflammatory cytokines and iNOS and NO production [67, 68].”

These studies were again mainly performed with immortalized BV-2 cells which transcriptionally look nothing like primary microglia (see Butovsky et al 2014 Nature Neuroscience vol 17(1)).  Be specific in the cell type used.

Reviewer 2 Report

1) I believe I fully understand what the author is trying to say. However, I have thought from the beginning that if we want to show the difference in TLR between tg338 and normal scrapie, we need to show the difference between tg338 homo and hetero and normal scrapie. Since there has been no comparison between the three groups since the beginning of the levy, it seems that the only issue is the expression of TLRs rather than the difference in TLRs between tg338 and normal scrapie.

Originally, the Tg338 scrapie model mouse differed from the classical scrapie and was positioned as an atypical scrapie. I image that it is very difficult to interpret the involvement of TLRs in the Tg388 model
